# Analysis of Psychological and Sleep Quality Characteristics of Young and Adult Para-Athletes with Cerebral Palsy During Competitive Period

**DOI:** 10.3390/sports13070203

**Published:** 2025-06-24

**Authors:** Fernando Muñoz-Hinrichsen, Felipe Herrera-Miranda, Sonny Riquelme, Matías Henríquez, Joel Álvarez-Ruf, María Isabel Cornejo, Luis Felipe Castelli Correia Campos

**Affiliations:** 1Laboratorio de Actividad Física, Salud, y Rendimiento Humano, Departamento de Kinesiología, Facultad de Artes y Educación Física, Universidad Metropolitana de Ciencias de la Educación, José Pedro Alessandri 774, Ñuñoa, Santiago 7760197, Chile; 2ParaLab, Departamento de Kinesiología, Facultad de Artes y Educación Física, Universidad Metropolitana de Ciencias de la Educación, José Pedro Alessandri 774, Ñuñoa, Santiago 7760197, Chile; joel.alvarez@umce.cl; 3Facultad de Ciencias de la Vida, Universidad Viña del Mar, Agua Santa 7055, Viña del Mar 2520000, Chile; fherrera@uvm.cl; 4Escuela de Ciencia del Deporte y Actividad Física, Facultad de Salud, Universidad Santo Tomás, José Miguel Carrera 158, Santiago 8320000, Chile; sonny.riquelme@inrpac.cl; 5Escuela de Kinesiología, Facultad de Salud, Universidad Santo Tomás, José Miguel Carrera 158, Santiago 8320000, Chile; mhenriquez24@santotomas.cl (M.H.); isa.cornejo27@gmail.com (M.I.C.); 6Laboratorio de Cognición y Comportamiento Sensoriomotor, Departamento de Kinesiología, Facultad de Artes y Educación Física, Universidad Metropolitana de Ciencias de la Educación, José Pedro Alessandri 774, Ñuñoa, Santiago 7760197, Chile; 7Laboratorio de Biomecánica Clínica, Carrera de Kinesiología, Facultad de Medicina Clínica Alemana de Santiago, Universidad del Desarrollo, Av. Plaza 680, Las Condes, Santiago 7610658, Chile; 8Núcleo de Investigación en Ciencias de la Motricidad Humana, Universidad Adventista de Chile, Camino a Las Mariposas 11771, Chillan 3780000, Chile; luisfelipe@cbdv.org.br

**Keywords:** football, paralympic, cerebral palsy, disability

## Abstract

Emotional processes and sleep quality have become fundamental aspects of performance in Paralympic sports among elite and youth athletes. The objective of this study was to compare levels of depression, stress, anxiety, and sleep quality among youth and adult athletes with cerebral palsy (CP) belonging to the national CP Football team in Chile. A total of 10 adult and 12 youth national team athletes participated, completing the DASS-21, Pittsburgh Sleep Quality Index, and Epworth Sleepiness Scale questionnaires. The athletes were competing in their respective categories at the 2024 Parapan American Games. A cross-sectional design was used to compare the parameters of depression, stress, anxiety, and sleep quality of youth and adult male athletes with CP of a national team selected to compete in a regional event. Significant differences were found where young athletes had lower levels of depression (χ^2^ = 4.77, *p* = 0.02, OR = 11.0) and anxiety (χ^2^ = 6.71, *p* = 0.01, OR = 16.5). Similar differences could be observed in favor of young athletes in sleep latency (*p* = 0.04; *d* = 0.34), bedtime (*p* = 0.02; *d* = 0.20), total hours of sleep (*p* = 0.04; *d* = 0.10), subjective sleep quality (*p* = 0.002; *d* = 0.56), and objective sleep quality (*p* < 0.001; *d* = 0.65). This study suggests that adult para-athletes from a national CP Football team exhibit higher levels of depression and anxiety compared to their youth counterparts. Additionally, objective and subjective measures show that adults experience poorer sleep quality. These findings highlight the need for targeted interventions by psychological support teams, aiming to enhance athlete performance by promoting healthy habits that address these mental health challenges.

## 1. Introduction

Studies related to emotional processes in the athlete population, particularly concerning depression, stress, and anxiety, have gained notable attention in recent years due to their crucial relevance and significant impact on sports performance, both in the Olympic and Paralympic arenas [1,2,3]. Inadequate emotion management can become a risk factor depending on an individual’s personal and sporting conditions and circumstances [4]. These factors may contribute to situations that trigger musculoskeletal complications, which can significantly stress athletes and often lead to physical pain, reduced sports performance, and time away from competition [5,6]. Moreover, the substantial psychological pressure to succeed [4] and the physical and emotional demands of training can become significant stressors, especially when these challenges are combined with additional responsibilities, such as school or work [7]. Similarly, the environment in which an athlete develops can significantly impact their mental health, and some athletes may exhibit perfectionist tendencies, setting high standards for themselves and demanding exceptional performance [8,9]. According to the data collected, 13.7% of the Chilean population suffers from depression, with the highest percentage among young people between 18 and 25 years old at 28.9%. Anxiety accounts for 25%, and stress accounts for 48% [10]. All of this makes it a significant health issue to consider.

Furthermore, research suggests that prioritizing the quality of sleep and rest is crucial for effective emotional management when planning the preparation and training of athletes at all levels, highlighting its influence on relevant factors contributing to sports performance [11,12]. Muscle repair and growth occur during sleep, particularly in the deep sleep stages, a process crucial for regulating hormones associated with muscle growth, metabolism, and stress, including growth hormone and cortisol. Sleep deprivation can impair cognitive function, affecting concentration, decision-making, and coordination, which results in slower reaction times, reduced accuracy in technical movements, and poor strategic planning during competition [11,13]. Previous studies have shown that sleep and rest are integral to athletic training, preparation, and performance during competition [14,15].

Cerebral palsy football is a sport specifically designed for individuals with congenital or acquired brain injuries, including impairments such as spasticity, ataxia, or dyskinesia [16,17], which is globally regulated by the International Federation of CP Football (IFCPF). This modality involves seven participants on each side, playing on a field of smaller dimensions than the standard (i.e., 70 × 50 m) with smaller goals (i.e., 5 × 2 m) and an offside rule not applied [18]. Footballers in this discipline exhibit physical and physiological characteristics that influence their motor performance in sports with intermittent demands, similar to those of their able-bodied counterparts. This modality involves a competitive cycle that includes young and adult athletes participating in world and regional circuits with the highest standards, exposing them to the psychological demands of high performance.

Consequently, physical, physiological, and psychological differences between young people and adults are relevant during training and competition. Young people are in a crucial stage of development, where their systems continue to grow, and may experience significant emotional and social instability in a sports competition context [19,20]. At this stage, sleep disorders are becoming increasingly common and may contribute to the development of stress and depression. The data observed in Chile indicate differences between the two populations, which contextually leads us to consider a comparison between the two groups [10]. Individuals with disabilities, particularly those with neurodevelopmental disorders, may face more significant challenges in falling asleep compared to the non-disabled population [21]. In CP football specifically, data are still limited. Still, qualitative studies [22] have shown experiences of competitive pressure, perceived inequity in functional classification, and difficulties in maintaining a balance between the demands of sports and daily life. This poses great relevance to this type of study, since it is a poorly researched topic. Regarding the characteristics observed in people with cerebral palsy, it was observed in the study developed by Ingrid Honna and collaborators (REF) that 33% of the participants reported symptoms of depression from moderate to extremely severe, and, regarding anxiety, 60% presented symptoms in the ranges of moderate to extremely severe, while 26% showed high levels of stress using the DASS-21 instrument [23]. Therefore, this study aimed to compare levels of depression, stress, anxiety, and sleep quality between youth and adult athletes with CP belonging to national para-football teams. This is relevant considering that both teams compete internationally, and these variables could affect their performance. We hypothesize that the group of young athletes will present better indicators in their sleep quality characteristics and psychological factors.

## 2. Materials and Methods

### 2.1. Participants

A cross-sectional design was used to compare parameters of depression, stress, anxiety, and sleep quality of youth and adult male athletes with CP of a national team selected to compete in a regional event (i.e., youth and adult Parapan American Games). The sample was selected using a non-probabilistic convenience method. It included athletes from national CP football teams in the ‘Youth’ categories who participated in the 2023 Youth Parapan American Games and were under 21 years of age (*n* = 12, 17.66 ± 2.01 years), and adults competing in the 2023 Parapan American Games (*n* = 10; 29.20 ± 3.85 years) (Table 1). This study’s objectives and protocols were explained to all participants who participated voluntarily. Informed consent was obtained from all participants, with minors requiring the permission of their guardians. All procedures received approval from the Diego Portales University Ethics Committee (Code 18-2024). Of the participants in the “Youth” category, 83.33% (*n* = 10) attend school or high school during their regular school day, and 16.7% (*n* = 2) are unemployed. In the “Adult” category, 100% (*n* = 10) work full-time or part-time. Both groups were evaluated while participating in their respective Parapan American Games. They were in the pre-event training camp, which took place seven days before their competition. Their training load consisted of two shifts, each with three hours of mixed training (technical and tactical), to prepare for the tournament. All members of both teams were evaluated; 100% of the athletes participated in the “Youth” category (12 out of 12 athletes), and 83.3% participated in the “Adult” group (10 out of 12 athletes, of whom two did not wish to respond voluntarily).

### 2.2. Procedures

Participants completed various assessments to evaluate psychological aspects and sleep quality during their training sessions, one week before each competitive event. The questionnaires were sent to them via text message and completed online. All questionnaires have been validated and previously used in Chile. The Spanish version of the Depression, Anxiety, and Stress Scale (DASS-21) questionnaire was used to rate aspects of mental health in the study participants [24,25]. This questionnaire consists of 21 items, each rated on a Likert scale. The scale ranges from 0 to 3, with the following meanings: 0 indicates ‘Did not apply to me at all’; 1 corresponds to ‘Applied to me to some degree, or some of the time’; 2 signifies ‘Applied to me to a considerable degree, or a good part of the time’; and 3 represents ‘Applied to me very much, or most of the time’. The items are grouped into three subscales, which are analyzed after a correction process that reorganizes them as follows: Depression (items 3, 5, 10, 13, 16, 17, and 21), Anxiety (items 2, 4, 7, 9, 15, 19, and 20), and Stress (items 1, 6, 8, 11, 12, 14, and 18). To evaluate each subscale independently, the scores of the corresponding items are summed and interpreted as follows: Depression: 5–6 indicates mild depression, 7–10 moderate, 11–13 severe, and 14 or more extremely severe; Anxiety: 4 indicates mild anxiety, 5–7 moderate, 8–9 severe; Stress: 10–12 moderate, 13–16 severe, and 17 or more extremely severe [25].

Regarding sleep quality, the Spanish version of the “Pittsburg Sleep Quality Index” was used, a direct and self-applied evaluation with questions corresponding to the factors involved in sleep quality [26,27]. This questionnaire includes 19 questions designed to evaluate seven dimensions of sleep quality. These dimensions are subjective sleep quality, which refers to how well or poorly individuals perceive their sleep; sleep latency, or the time it takes to fall asleep; sleep duration, measured by the number of hours slept; sleep efficiency, calculated as the percentage of time spent sleeping relative to the time spent in bed; sleep disturbances, encompassing symptoms such as insomnia, apnea, and difficulties in maintaining sleep; use of sleep medications; and daytime dysfunction, which involves feelings of sleepiness during the day and the potential challenges that may arise from it [28].

Sleepiness was analyzed with the Spanish version of the “Epworth Sleepiness Scale” [29]. This questionnaire asks, “How likely are you to feel drowsy, nod off, or fall asleep in the following situations instead of just feeling tired? The instrument uses the following scale for each situation: 0 = No likelihood of nodding off or falling asleep; 1 = Low likelihood of nodding off or falling asleep; 2 = Moderate likelihood of nodding off or falling asleep; and 3 = High likelihood of nodding off or falling asleep. A total of 8 questions totaling 24 points, where 1–6: normal, 7–8: average sleepiness, and 9–24: abnormal (i.e., possibly pathological) sleepiness.

### 2.3. Statistical Analysis

Descriptive statistics were employed to summarize the results from each questionnaire, including frequencies, means, standard deviations, and percentages where appropriate. To assess the distribution of the data, the Shapiro–Wilk test (for sample sizes n < 50) was employed, which confirmed the suitability of parametric statistical methods. Subsequently, parametric tests were used to analyze variables related to Depression, Anxiety, and Stress. For the DASS-21 test, the Chi-Square test of proportions (χ^2^) was used to evaluate the balance of response proportions and calculate the Odds Ratio (OR) with a 95% confidence interval (CI). Sleepiness and quality were examined through Sleep Latency, Sleep Quantity, and Sleep Quality metrics. The student’s *t*-test was performed to compare means and calculated effect sizes (d) with thresholds for interpretation: >0.80 indicates a large effect, >0.50 and <0.79 a moderate effect, >0.25 and <0.49 a small effect, and <0.25 a trivial effect [30]. A significant level of *p* < 0.05 was set for all tests. The analysis used Jamovi “The Jamovi project 2022 2.3.28” and Microsoft Excel 365 [31].

## 3. Results

Analysis of the DASS-21 scale scores revealed that 8.3% of youth-level athletes exhibited symptoms of depression. In contrast, 50.0% of adult athletes experienced depression, primarily at a moderate level, indicating a significant difference between the two groups (χ^2^ = 4.77, *p* = 0.02, OR = 11.0). A similar pattern is observed in the “Anxiety” condition, where 8.3% of youth athletes are affected compared to a significantly higher 60.0% in adults (χ^2^ = 6.71, *p* = 0.01, OR = 16.5). Although the differences in the “Stress” condition are not statistically significant, adults still exhibit a higher prevalence at 50.0%, compared to 16.7% in youth (Table 2 and Table 3).

Significant differences in sleep quality were observed between the groups, particularly in sleep latency, where adult athletes had a longer latency of 35.64 ± 27.81 min (*p* = 0.04; *d* = 0.34, small). Additionally, there were significant differences in bedtime (*p* = 0.02; *d* = 0.20, trivial), with adults going to bed later, at an average time of 23:24. This later bedtime affected the total hours of sleep, with the adult group averaging 6.72 ± 1.34 h, significantly less (*p* = 0.04; *d* = 0.10, trivial) than the youth team, which averaged 7.75 ± 0.75 h of sleep (Table 4). In terms of sleep quality, qualitative analysis reveals that youth participants perceive their sleep as ‘Quite bad’ (2.41 ± 0.51), while adults rate theirs as ‘Quite good’ (1.54 ± 0.52), reporting significant differences (*p* = 0.002; *d* = 0.56, moderate). However, this contrasts with the objective measurements of sleep quality, where youth athletes fall closer to the ‘Fairly good’ range (5.91 ± 1.37), whereas adults remain in the ‘Good’ category (8.54 ± 2.29), with a more pronounced significant difference (*p* < 0.001; *d* = 0.65, moderate). Regarding sleepiness, both groups were categorized as ‘Normal’ (Table 4).

## 4. Discussion

This study aimed to compare the levels of depression, stress, anxiety, and sleep quality between youth and adult para-athletes with CP who are members of national para-football teams. The results revealed a significantly higher proportion of anxiety among adult team members (60%) compared to their youth counterparts (8.3%). This finding contrasts with what was observed in adult athletes, where anxiety symptoms were lower in more experienced individuals [9], possibly because they tend to develop more advanced coping mechanisms over their careers to manage the challenges of high-level competition [30]. In addition, younger athletes may require additional developmental-specific support to progress these skills due to their immaturity or limited exposure to such situations [32].

Youth athletes may experience heightened anxiety due to several factors, including the pressure to perform in competition, expectations from coaches and parents, whether conveyed implicitly or explicitly, and the challenge of balancing sports with other aspects of their lives, such as academics, leisure activities, and daily routines [19]. Similarly, adult para-athletes may report higher anxiety levels, potentially linked to the complexities of managing their disability, the pressure to represent a marginalized group, and the need to prove their capabilities despite facing social and physical barriers [33,34]. This situation is repeated when observing the traits linked to the presence of depression in the adult group versus the youth group (50% versus 8.3%, respectively). This difference is likely influenced by the unique risk factors adult athletes face, such as the pressure to maintain elite-level performance, the challenges of aging, uncertainty about life after sports, and the expectations of sponsors and media, which are factors that could affect mental health [35]. Additionally, another relevant issue that may be a factor to highlight is career transitions, such as retirement from sports, which can be a high-risk period for depression due to the loss of identity, daily structure, and social support that sport provides [36].

Furthermore, a previous study on Australian para-athletes reported lower self-esteem than non-para-athletes, highlighting a factor that should be considered when addressing this population’s specific needs for mental health [37]. Similarly, Meidl et al. [2] found high rates of mental health issues among German elite para-athletes, emphasizing the need for continued monitoring to enable early detection and appropriate intervention. Effective interventions for both groups include access to mental health services, education about depression, and creating a sports environment that encourages mental well-being, especially considering the particularities of athletes with a disability [38].

By analyzing sleep hours, it is possible to describe two relevant issues in the different groups of athletes with a disability. First, the results indicated that adult athletes (6.72 ± 1.34 h) do not meet the standard of hours of sleep recommended by the literature, and youth athletes (7.75 ± 0.75 h) remain on the limit, which indicates that adult athletes should get ~7 h per night [39]. However, some studies suggest that adults can benefit from up to 7 to 8 h of sleep to maximize recovery. In young individuals, the recommendation is to sleep between 8 and 10 h per night since adolescents have more significant sleep needs due to their stage of development, and the combination of intense training with their academic demands can increase these requirements [40,41]. The present results are similar to those of Duran Agüero et al. [42], who studied a Chilean sample of para-athletes and described low sleep quality and insomnia, factors that can influence performance. Moreover, early studies found a high prevalence of sleep disorders and an association between the severity of health problems and sleep quality, reflecting the complications faced by athletes with a disability [43]. The specific health characteristics of athletes with CP, such as spasticity, involuntary movements, or muscle spasms, may impact sleep duration and quality, factors that should be addressed to achieve improvements [43]. The second point to analyze is that there are significant differences in favor of youth category athletes regarding hours of sleep, which coincides with the recommendations regarding differences according to age [12]. Given the importance of sleep for development and athletic performance, strategies to optimize sleep should be tailored to the specific health conditions that may affect it in young athletes with cerebral palsy [41].

The competitive demands placed on para-athletes with cerebral palsy (CP) are multifactorial and extend beyond the immediate context of performance. Participation at the international level, such as the Parapan American Games, introduces unique psychological, physical, and logistical stressors. Athletes must adhere to intense training schedules, undergo rigorous classification systems, and adapt to environments that may not always be inclusive or accommodating to their needs. These pressures are amplified in para-sport due to limited resources, fewer competitive opportunities, and reduced public visibility compared to able-bodied counterparts, which may contribute to increased psychological strain. Studies suggest that the psychological burden of representing a minority group in elite sport contexts can exacerbate performance anxiety, especially when combined with disability-related stigmas or a perceived need to validate one’s legitimacy as an athlete in mainstream sporting culture [1,9,37]. Moreover, para-athletes often face unique biomechanical and physiological challenges that require customized coaching and recovery protocols, which can be inconsistently implemented across national systems. Inconsistency in access to high-quality support staff, sports psychologists, and adapted facilities has also been associated with increased perceived stress in Paralympic athletes [2,38].

Another dimension of competition-related stress involves the psychological expectations associated with the athlete’s competitive identity. Athletes may internalize the pressure to achieve not only for personal gain but also to contribute to broader narratives around disability and excellence. This can result in perfectionistic tendencies and fear of failure, which have been strongly associated with anxiety and depressive symptoms in both able-bodied and disabled athletes [4,5,8]. These demands are often intensified during key events, where performance outcomes may determine ongoing funding, sponsorship, or selection for future international competition. Furthermore, many para-athletes experience prolonged travel, disrupted routines, and extended time away from their support networks during international tournaments. These contextual factors can significantly disrupt sleep patterns and reduce psychological recovery time, especially when time-zone changes and pre-competition arousal interfere with the circadian rhythm [12,13]. The added burden of performance evaluation, media exposure, and expectations from coaches or national federations during these events can significantly increase cortisol levels and reduce sleep efficiency [11,14]. Collectively, these elements suggest that competitive periods are not only physically demanding but also represent a high-risk window for deteriorating mental health and sleep quality. Given these findings, tailored interventions that support psychological resilience and regulate sleep during high-demand competition phases should be a key consideration for sports science and health professionals working with CP athletes.

In addition to competitive demands, the intersection between athletic engagement and personal responsibilities, such as school or employment, plays a critical role in shaping mental health outcomes in para-athletes. The findings of the present study indicate that 83.3% of youth athletes are enrolled in school, while 100% of adults are employed either full-time or part-time. These parallel obligations may significantly contribute to elevated levels of anxiety, stress, and disrupted sleep patterns, particularly when time and energy must be distributed across multiple high-demand domains. Adolescents balancing academic workloads with training commitments often experience reduced sleep duration and quality, leading to increased fatigue and impaired cognitive performance. This is particularly relevant in youth with CP, who may require more time to complete academic tasks due to motor or mental impairments, thereby extending their daily workload beyond that of their peers. Similarly, adult athletes with disabilities may face job insecurity or the need to accommodate medical appointments or therapy within standard working hours, compounding their physical and emotional fatigue [33,36].

The dual role of being both an elite athlete and a student or worker creates a context of chronic stress that may be underestimated in traditional sports psychology frameworks. Research has shown that time scarcity and role conflict are significant predictors of sleep disturbances and psychological distress in athlete populations [38,39]. Among para-athletes, these pressures may be intensified by the additional time required for self-care routines, mobility, and transportation, which are often overlooked in planning schedules or support programs. Furthermore, both academic and work environments may not always be accommodating or flexible, resulting in the accumulation of unresolved stressors [41]. For example, an adult para-athlete who must train in the early morning hours before work and then return to a job requiring physical effort may experience a compounded load on both neuromuscular and emotional systems, reducing opportunities for recovery and regeneration [16]. The relationship between stress, anxiety, and poor sleep is bidirectional, meaning that poor sleep quality can further increase emotional dysregulation and reduce coping capacity [10,44]. This dynamic cycle highlights the importance of comprehensive mental health care and sleep management strategies that consider the entirety of the athlete’s life context. Practitioners should consider implementing structured support systems that help athletes develop time-management skills, establish realistic training loads, and advocate for flexibility in academic or occupational settings. Such holistic approaches are likely to mitigate mental health risks and optimize both performance and well-being across developmental stages [45,46].

This study has certain limitations, particularly the specificity of the participant sample, which introduces complexity when making comparisons. Additionally, the participants were evaluated during different tournaments, corresponding to their respective categories, which may have introduced variability in their responses due to differing contextual factors. However, the strength of this research lies in its novelty within the region, providing data on national team athletes contextualized to the reality of a Latin American country. The high percentage of national team members in both categories addressing this issue also suggests it is a phenomenon of relevance for this population. Other limitations are that this study is cross-sectional, so it is recommended to conduct longitudinal studies that allow for monitoring the processes of these groups of athletes. Finally, while the sample of participants represents the majority of national team players in both groups, it would be beneficial to analyze athletes from clubs and other teams. Future studies could investigate the relationships between physiological variables that influence emotional states and sleep quality, as well as the potential impact of medication use, spasticity, or hypertonia on sleep patterns. In addition, studying women para-athletes in this sports discipline may allow for a deeper understanding of these factors that influence overall well-being. The results have practical implications for coaches and healthcare staff, highlighting the importance of monitoring psychological factors such as depression, anxiety, stress, and sleep duration, while considering age groups, health-specific factors, and the competitive season.

## 5. Conclusions

This study suggests that adult para-athletes from a national CP Football team exhibit higher levels of depression and anxiety compared to their youth counterparts. Additionally, both objective and subjective measures show that adults experience poorer sleep quality. These findings highlight the need for targeted interventions by psychological support teams, aiming to enhance athlete performance by promoting healthy habits that address these mental health challenges.

## Figures and Tables

**Table 1 sports-13-00203-t001:** General characteristics of athletes with cerebral palsy from a national football team.

Athlete	Category	Age (Years)	Sport Classes	Game Position
1	Youth	21	2	Goalkeeper
2	Youth	16	2	Defender
3	Youth	16	2	Defender
4	Youth	18	2	Defender
5	Youth	19	2	Defender
6	Youth	15	2	Midfielder
7	Youth	15	2	Midfielder
8	Youth	19	2	Midfielder
9	Youth	19	2	Midfielder
10	Youth	20	2	Midfielder
11	Youth	16	1	Striker
12	Youth	18	1	Striker
13	Adult	30	2	Goalkeeper
14	Adult	30	1	Goalkeeper
15	Adult	34	1	Goalkeeper
16	Adult	30	2	Defender
17	Adult	32	2	Defender
18	Adult	33	2	Defender
19	Adult	23	2	Midfielder
20	Adult	25	2	Midfielder
21	Adult	31	3	Midfielder
22	Adult	24	2	Striker

**Table 2 sports-13-00203-t002:** Description of the degrees of depression, anxiety, and stress based on the DASS-21 questionnaire of athletes with cerebral palsy from a national team.

Variables		Characteristic		
Athletes	Level	Depression	Anxiety	Stress
1	Youth	0 (No depression)	0 (No anxiety)	2 (No stress)
2	Youth	0 (No depression)	0 (No anxiety)	1 (No stress)
3	Youth	1 (No depression)	1 (No anxiety)	1 (No stress)
4	Youth	0 (No depression)	0 (No anxiety)	3 (No stress)
5	Youth	0 (No depression)	0 (No anxiety)	2 (No stress)
6	Youth	1 (No depression)	1 (No anxiety)	6 (No stress)
7	Youth	1 (No depression)	1 (No anxiety)	9 (Slight)
8	Youth	1 (No depression)	3 (No anxiety)	4 (No stress)
9	Youth	3 (No depression)	3 (No anxiety)	5 (No stress)
10	Youth	3 (No depression)	1 (No anxiety)	1 (No stress)
11	Youth	5 (Slight)	8 (Severe)	10 (Moderate)
12	Youth	1 (No depression)	2 (No anxiety)	4 (No stress)
13	Adult	9 (Moderate)	4 (Slight)	8 (Slight)
14	Adult	3 (No depression)	4 (Slight)	7 (No stress)
15	Adult	8 (Moderate)	8 (Severe)	10 (Moderate)
16	Adult	10 (Moderate)	5 (Moderate)	7 (No stress)
17	Adult	2 (No depression)	3 (No anxiety)	8 (Slight)
18	Adult	0 (No depression)	1 (No anxiety)	3 (No stress)
19	Adult	2 (No depression)	1 (No anxiety)	0 (No stress)
20	Adult	8 (Moderate)	9 (Severe)	11 (Moderate)
21	Adult	0 (No depression)	0 (No anxiety)	5 (No stress)
22	Adult	10 (Moderate)	11 (Extreme)	13 (Severe)

Note: Data expressed in the categorization of the DASS-21 instrument.

**Table 3 sports-13-00203-t003:** Percentages and odds ratio based on DASS-21 of athletes with cerebral palsy from a national team.

	Category			χ^2^	*p*	OR	IC 95%
	Youth	Adults	Total				
No Depression	91.7%	50.0%	72.7%	4.77	0.02 *	11.0	1.00–120
Depression	8.3%	50.0%	27.3%				
No Anxiety	91.7%	40.0%	68.2%	6.71	0.01 *	16.5	1.49–183
Anxiety	8.3%	60.0%	31.8%				
No Stress	83.3%	50.0%	68.2%	2.79	0.09	5.00	0.70–35.5
Stress	16.7%	50.0%	31.8%				

Note: *p* < 0.05 *, percentages based on column per category (Depression, Anxiety, Stress). χ^2^ = Chi-Square, OR = Odds Ratio, IC = Confidence interval.

**Table 4 sports-13-00203-t004:** Sleep quality and sleepiness of athletes with cerebral palsy from a national team.

Variable	Category	Mean (SD)	Min–Max	*p* (*d*)
Sleep latency (min)	Youth	14.75 ± 12.50	2.00–45.00	0.04 * (0.34)
	Adult	35.64 ± 27.81	2.00–90.00	
Bedtime (hours)	Youth	22:17 ± 0.62	21:00–23:00	0.02 * (0.20)
	Adult	23.24 ± 1.27	21:30–02:00	
Wake up time (hours)	Youth	7:22 ± 1.75	5:00–12:20	0.96 (0.03)
	Adult	6.45 ± 1.05	5:00–9:00	
Amount of sleep (hours)	Youth	7.75 ± 0.75	7.00–9.00	0.04 * (0.10)
	Adult	6.72 ± 1.34	5:00–9:00	
Qualitative sleep quality (Likert score)	Youth	2.41 ± 0.51	2.00–3.00	0.002 * (0.56)
	Adult	1.54 ± 0.52	1.00–2.00	
Quantitative sleep quality (Score)	Youth	5.91 ± 1.37	3.00–7.00	<0.001 * (0.65)
	Adult	8.54 ± 2.29	6.00–14.00	
Sleepiness (Score)	Youth	1.08 ± 0.28	1.00–2.00	0.017 * (0.49)
	Adult	1.91 ± 0.94	1.00–3.00	

Note: *p* < 0.05 *; Student’s *t*-test, *d* = Effect size.

## Data Availability

All research data is available upon request at fernando.munoz_h@umce.cl.

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
