# Peer review of "Analysis of Psychological and Sleep Quality Characteristics of Young and Adult Para-Athletes with Cerebral Palsy During Competitive Period"

_sports, 2025, doi:10.3390/sports13070203_

Round 1
Reviewer 1 Report
Comments and Suggestions for Authors General CommentsThis manuscript addresses a relevant and novel topic by comparing levels of depression, anxiety, stress, and sleep quality in youth versus adult para-athletes with cerebral palsy. It is particularly valuable given the national-level status of participants and the regional context (Chile), where such data are scarce.
While the overall design and methodology are sound, several key areas require clarification or expansion, particularly concerning measurement terminology, contextual background of participants, control of confounding variables, and justification of sample size.Major Points for Revision:
Clarification of Hypothesis and Study Purpose
Please clearly state the hypothesis and research objective in the introduction. What were the authors expecting to find? Why was it important to compare youth and adult CP athletes?
Ambiguous Use of Measurement Terms
Terms such as “subjective sleep quality” and “objective sleep quality” should be defined precisely. PSQI is a self-reported instrument, so “objective” may not be an appropriate label.
Background Information on Participants
The authors should include details such as employment/school status or other contextual factors that could impact sleep and mental health.
Control of Confounding Variables
Important variables such as sleep medication use, level of impairment, training volume, and the timing of questionnaire administration relative to competition are not discussed.
Lack of Sample Size Justification
Please include a brief comment on the adequacy of the sample size or a post-hoc power analysis, especially considering the small number of participants (n = 22).
Discussion Over-Interpretation
Some parts of the discussion rely on speculative reasoning or assumptions not directly supported by the data. A more data-driven interpretation would strengthen the manuscript.Minor Points:
The classification of 6.7 hours of sleep as "short" could be reconsidered, as it lies within the normal adult range.
Please correct typographical inconsistencies (e.g., “Estress” should be “Stress”).
Recommendation:
I recommend Minor Revision, assuming the issues above are adequately addressed. However, if major points remain unresolved, reclassification as Major Revision may be warranted. Comments on the Quality of English LanguageThe manuscript is generally well written and clear. Only minor improvements to phrasing or terminology may be needed (e.g., the use of “objective” for self-reported sleep quality).
Author Response
Comment 1:
General Comments
This manuscript addresses a relevant and novel topic by comparing levels of depression, anxiety, stress, and sleep quality in youth versus adult para-athletes with cerebral palsy. It is particularly valuable given the national-level status of participants and the regional context (Chile), where such data are scarce.
Response 1:
Thank you very much for your comments. We hope the study we conducted will contribute to the development of Paralympic sports in Chile, particularly in the psychological management of athletes.
Comment 2:
While the overall design and methodology are sound, several key areas require clarification or expansion, particularly concerning measurement terminology, contextual background of participants, control of confounding variables, and justification of sample size.
Response 2:
We find your comment very important. We hope the adjustments presented below will meet your requirements.
Major Points for Revision:
Comment 3:
Clarification of Hypothesis and Study Purpose
Please clearly state the hypothesis and research objective in the introduction. What were the authors expecting to find? Why was it important to compare youth and adult CP athletes?
Response 3:
We consider this comment to be of great importance. We've adjusted the text appearing between rows 88-91 to address this issue: This is relevant considering that both teams compete internationally, and these variables could affect their performance. We hypothesize that the group of young athletes presents better indicators in their sleep quality characteristics and psychological factors.
Comment 4:
Ambiguous Use of Measurement Terms
Terms such as “subjective sleep quality” and “objective sleep quality” should be defined precisely. PSQI is a self-reported instrument, so “objective” may not be an appropriate label.
Response 4:
We consider the comment very important. We made the change in the document, replacing the terms "objective" with "quantitative" and "subjective" with "qualitative."
Comment 5:
Background Information on Participants
The authors should include details such as employment/school status or other contextual factors that could impact sleep and mental health.
Response 5:
We consider this comment to be significant, so we have added a paragraph in the "Participants" section to explore this topic in more depth:
Of the participants in the "Youth" category, 83.33% (n = 10) attend school or high school during their regular school day, and 16.7% (n = 2) are unemployed. In the "Adult" category, 100% (n = 10) work full-time or part-time.
Comment 6:
Control of Confounding Variables
Important variables such as sleep medication use, level of impairment, training volume, and the timing of questionnaire administration relative to competition are not discussed.
Response 6:
This comment is significant; unfortunately, we don't have data on the use of sleeping pills or medications. We're considering amending it in the text as follows:
Both groups were evaluated while participating in their respective Parapan American Games. They were in the pre-event training camp, which took place seven days before their competition. Their training load consisted of two shifts, each with three hours of mixed training (technical and tactical), to prepare for the tournament.
Comment 7:
Lack of Sample Size Justification
Please include a brief comment on the adequacy of the sample size or a post-hoc power analysis, especially considering the small number of participants (n = 22).
Response 7:
The comment is very important, so we considered modifying it in the manuscript:
All members of both teams were evaluated; 100% of the athletes participated in the "Youth" category (12 out of 12 athletes), and 83.3% participated in the "Adult" group (10 out of 12 athletes, of whom two did not wish to respond voluntarily).
Comment 8:
Discussion Over-Interpretation
Some parts of the discussion rely on speculative reasoning or assumptions not directly supported by the data. A more data-driven interpretation would strengthen the manuscript.
Response 8
We find your comment very important. We hope the adjustments presented in the manuscript meet your requirements.
Comment 9:
Minor Points:
The classification of 6.7 hours of sleep as "short" could be reconsidered, as it lies within the normal adult range.
Please correct typographical inconsistencies (e.g., “Estress” should be “Stress”).
Response 9
Thank you very much for your comment. According to the literature, the average sleep time for adults is 7 to 8 hours. It's essential to maintain this range.

Reviewer 2 Report
Comments and Suggestions for Authors
The present work aimed to compare levels of depression, stress, anxiety, and sleep quality among youth and adult athletes with cerebral palsy belonging to the national CP Football team in Chile. And it was identified that adult para-athletes exhibit higher levels of depression and anxiety compared to their youth counterparts. Additionally, objective and subjective measures show that adults experience poorer sleep quality.
Despite the overall merit of this paper, there are some points that should be addressed.
The methodology section of the Abstract should be strengthened with more information.
The introduction section has relevant information; however, the prevalence of depression, anxiety, and stress in this population should be included. Despite the impact that it has on performance, it allows us to understand whether there is a problem with the study’s population.
The pertinence of the comparison between adults and youth should be better justified.
In the methodology section, the author should indicate whether the Spanish version of the scales is adequate to be used in Chile and in both populations (i.e., adult and youth).
In Table 3, the authors should identify OR and IC in the table’s notes.
The discussion section should include an analysis of other variables (e.g., the sport level enrollment; differences between competitive demand; other personal responsibilities; whether they are professionals, etc.).
Transversal study and non-representative sample should be indicated as study limitations.
Finally, I would like to thank the authors for their work and hope that my feedback can improve the quality of this manuscript.
Author Response
Comment 1:
The present work aimed to compare levels of depression, stress, anxiety, and sleep quality among youth and adult athletes with cerebral palsy belonging to the national CP Football team in Chile. And it was identified that adult para-athletes exhibit higher levels of depression and anxiety compared to their youth counterparts. Additionally, objective and subjective measures show that adults experience poorer sleep quality.
Despite the overall merit of this paper, there are some points that should be addressed.
Response 1
Thank you very much for your comments. We hope the study we conducted will contribute to the development of Paralympic sports in Chile, particularly in the psychological management of athletes.
Comment 2:
The methodology section of the Abstract should be strengthened with more information.
Response 2
Thank you very much for your comments. We've strengthened the methodology section, particularly in the "Participants" section, to provide a more detailed explanation.
Comment 3:
The introduction section has relevant information; however, the prevalence of depression, anxiety, and stress in this population should be included. Despite the impact that it has on performance, it allows us to understand whether there is a problem with the study’s population.
Response 3
This is a significant comment. We've added a paragraph in the introduction that explains what was requested:
According to the data collected, 13.7% of the Chilean population suffers from depression, with the highest percentage among young people between 18 and 25 years old at 28.9%. Anxiety accounts for 25%, and stress accounts for 48%. All of this makes it a significant health issue to consider.
Comment 4
The pertinence of the comparison between adults and youth should be better justified.
Response 4
This is a significant comment. We've added a paragraph in the introduction that explains what was requested:
The data observed in Chile indicate differences between the two populations, which contextually leads us to consider a comparison between the two groups.
Comment 5:
In the methodology section, the author should indicate whether the Spanish version of the scales is adequate to be used in Chile and in both populations (i.e., adult and youth).
In Table 3, the authors should identify OR and IC in the table’s notes.
Response 5
Thank you very much for your comment. Regarding the Chilean context, the manuscript was modified as follows:
All questionnaires have been validated and previously used in Chile.
Respecto de la table 3 se agregan en las notas de la table según lo indicado.
Comment 6:
The discussion section should include an analysis of other variables (e.g., the sport level enrollment; differences between competitive demand; other personal responsibilities; whether they are professionals, etc.).
Response 6
We find your comment very important. We hope the adjustments presented in the manuscript can meet your requirements.
Comment 7:
Transversal study and non-representative sample should be indicated as study limitations.
Response 7
Thank you very much for your comment. It has been modified in the manuscript and is presented as follows in the discussion section:
Other limitations are that the study is cross-sectional, so it is recommended to conduct longitudinal studies that allow for monitoring the processes of these groups of athletes. Finally, while the sample of participants represents the majority of national team players in both groups, it would be beneficial to analyze athletes from clubs and other teams.
Comment 8:
Finally, I would like to thank the authors for their work and hope that my feedback can improve the quality of this manuscript.
Response 8
We value your feedback very much. We hope the adjustments presented below will meet your needs.

Round 2
Reviewer 2 Report
Comments and Suggestions for Authors
Dear authors,
Thank you for addressing my comments.
However, regarding my previous comment nº2, I can not find the alterations made in Abstract. Also, I consider that my previous comment nº3 should be more explored. Despite the importance of the added information, if possible, the authors should specify this content to the study's population (i.e., Young and Adult Para-Athletes with Cerebral Palsy).
Best regards,
Author Response
Comment 1:
Comment nº2, I can not find the alterations made in Abstract. Also, I consider that my previous
Response 1:
We appreciate the comment and propose the following adjustment, considering an extension appropriate for what the journal stipulates:
“A cross-sectional design was used to compare the parameters of depression, stress, anxiety, and sleep quality of youth and adult male athletes with CP of a national team selected to compete in a regional event.”
Comment 2:
Comment nº3 should be more explored. Despite the importance of the added information, if possible, the authors should specify this content to the study's population (i.e., Young and Adult Para-Athletes with Cerebral Palsy).
Response 2:
We consider your comment very important. We have modified the manuscript with the following proposal:
“In CP football specifically, data are still limited. Still, qualitative studies have shown experiences of competitive pressure, perceived inequity in functional classification, and difficulties in maintaining a balance between the demands of sports and daily life. This poses great relevance to this type of study, since it is a poorly researched topic. Regarding the characteristics observed in people with cerebral palsy, Depression, it was observed in the study developed by Ingrid Honna and collaborators (REF), that 33% of the participants reported symptoms of depression from moderate to extremely severe, regarding anxiety, 60% presented symptoms in the ranges of moderate to extremely severe, and stress 26% showed high levels of stress using the DASS-21 instrument”.
References:
Sivaratnam, C.; Howells, K.; Stefanac, N.; Reynolds, K.; Rinehart, N. Parent and Clinician Perspectives on the Participation of Children with Cerebral Palsy in Community-Based Football: A Qualitative Exploration in a Regional Setting. Int. J. Environ. Res. Public Health 2020, 17, 1102. https://doi.org/10.3390/ijerph17031102
Honan, I.; Waight, E.; Bratel, J.; Given, F.; Badawi, N.; McIntyre, S.; Smithers-Sheedy, H. Emotion Regulation Is Associated with Anxiety, Depression and Stress in Adults with Cerebral Palsy. J. Clin. Med. 2023, 12, 2527. https://doi.org/10.3390/jcm12072527
